# Chemical Diversity of Metal Sulfide Minerals and Its Implications for the Origin of Life

**DOI:** 10.3390/life8040046

**Published:** 2018-10-10

**Authors:** Yamei Li, Norio Kitadai, Ryuhei Nakamura

**Affiliations:** 1Earth-Life Science Institute, Tokyo Institute of Technology, 2-12-1 Ookayama, Meguro-ku, Tokyo 152-8550, Japan; nkitadai@elsi.jp; 2Biofunctional Catalyst Research Team, RIKEN Center for Sustainable Resource Science, 2-1 Hirosawa, Wako, Saitama 351-0198, Japan

**Keywords:** origin of life, prebiotic chemistry, mineral catalysis, sulfide minerals, mineral diversity, density functional theory, electrocatalysis

## Abstract

Prebiotic organic synthesis catalyzed by Earth-abundant metal sulfides is a key process for understanding the evolution of biochemistry from inorganic molecules, yet the catalytic functions of sulfides have remained poorly explored in the context of the origin of life. Past studies on prebiotic chemistry have mostly focused on a few types of metal sulfide catalysts, such as FeS or NiS, which form limited types of products with inferior activity and selectivity. To explore the potential of metal sulfides on catalyzing prebiotic chemical reactions, here, the chemical diversity (variations in chemical composition and phase structure) of 304 natural metal sulfide minerals in a mineralogy database was surveyed. Approaches to rationally predict the catalytic functions of metal sulfides are discussed based on advanced theories and analytical tools of electrocatalysis such as proton-coupled electron transfer, structural comparisons between enzymes and minerals, and in situ spectroscopy. To this end, we introduce a model of geoelectrochemistry driven prebiotic synthesis for chemical evolution, as it helps us to predict kinetics and selectivity of targeted prebiotic chemistry under “chemically messy conditions”. We expect that combining the data-mining of mineral databases with experimental methods, theories, and machine-learning approaches developed in the field of electrocatalysis will facilitate the prediction and verification of catalytic performance under a wide range of pH and Eh conditions, and will aid in the rational screening of mineral catalysts involved in the origin of life.

## 1. Introduction

The origin of life on Earth is generally envisioned as having started from abiotic syntheses of basic building blocks requisite for metabolism and replication [1,2,3,4,5]. Metal sulfides have been proposed as key players in these prebiotic processes in several scenarios, such as the “iron–sulfur world” by Wächtershäuser [6,7], the “iron–sulfur membrane model” by Russell and Hall [5,8,9] and more recently by Lane and Martin [10], the “zinc world” hypothesis by Mulkidjanian and Galperin [11,12], and the “geoelectrochemistry driven origin of life” by Nakamura, Yamamoto [13,14,15,16,17,18], and Barge [19,20,21]. Metal sulfides are ubiquitous in reducing environments, including sulfidic ores [22,23], deep-sea hydrothermal vent deposits [8,24,25,26], sulfide-rich euxinic sediment environments (e.g., black sea) [27,28,29], and black shale [30,31,32]. In deep-sea hydrothermal vent environments, where massive sulfide deposits are produced, many dissolved transition metals are concentrated in hydrothermal fluids mainly as chloride complexes [22,23]. The generally low solubility of metal sulfides possibly limited the bio-availability of metal ions in the early ocean and has led to the assumption that the availability of metal ions in the ocean may have constrained metabolic pathways in early life [33,34,35]. 

Deep-sea hydrothermal systems associated with the serpentinization of ultramafic rocks are among the most plausible geological settings for life to have originated [8,10,13,15,17,18,36,37]. The temperature, pH, and chemical composition differences between hydrothermal fluid and seawater generate a steep redox gradient across the sulfide-rich vent rocks, thereby serving as the driving force for prebiotic chemistry [5,8,9]. In addition, the generation of a chemical potential gradient across electrically conductive metal sulfides provides a continuous and unidirectional supply of high-energy electrons from the inside of the vent to the outside of the rock wall, as has been proposed based on the electrochemical analysis of hydrothermal vent minerals [17,19,21] and the electrochemical potential mapping of deep-sea hydrothermal fields [13,17,18]. In this environment, the high-energy electrons continuously supplied by the natural geoelectrochemical reactor trigger CO_2_ reduction to CO and CH_4_ and nitrate and nitrite reduction to NO, N_2_O, and NH_3_, as demonstrated in recent laboratory investigations [14,15,16,38]. Under the continuous flow of electrical current from a hot, reductive hydrothermal fluid to cold seawater, the ocean-vent interface is ideal for prebiotic chemistry, as organic molecules crucial for the life’s origin, including α-keto acids, amino acids, and oligonucleotides are typically unstable at high temperatures [5,8]. In addition, the unique nano- and micro-scale structures of metal sulfides in hydrothermal vent environments may have promoted the concentration and organization of the synthesized molecules, preventing them from diffusing into seawater [39,40].

The thermo- and electro-catalytic properties of metal sulfides are critical for understanding how the pH, temperature, and Eh disequilibria between hydrothermal fluids and seawater trigger prebiotic organic synthesis. In mineralogy, metal sulfides have been studied mainly with respect to crystallography, thermodynamic and high-pressure phase transition, air stability, electric and magnetic properties, trace element incorporation, and mineral conversion [41,42]. Their thermo-catalytic activities towards hydrogenation, hydrodesulfurization and hydrodenitrogenation have also been studied under gas-phase, high temperature conditions in petrochemistry since the 1920s [43,44]. However, the thermal and electrochemical catalysis of metal sulfides remain poorly explored in the context of the origin of life, particularly under simulated hydrothermal vent conditions. 

To synthesize macromolecules (phospholipids, oligo-nucleotides, and peptides) and their respective building blocks (fatty acids, glycerol, ribose, nucleobases, nucleotides, and amino acids) from simple molecules (CO_2_, N_2_, NO_3_^−^/NO_2_^−^, and H_2_) to form a complex prebiotic chemical network, efficient mineral catalysts must be identified, which requires extensive screening efforts. Although several studies have used sulfides as catalysts for abiotic carbon and nitrogen fixation and peptide bond formation [14,15,16,38,45,46,47,48,49,50], limited types of organic products were typically formed and the reported activities and selectivity were generally much lower than those found in contemporary biological systems. These problems suggest that new types of earth-abundant mineral catalysts that function efficiently under geologically relevant conditions are needed for prebiotic synthesis. 

Numerous minerals are identified in natural environments yearly, contributing to a huge library of mineral catalysts, and the number of unknown minerals awaiting discovery is predicted to be even larger [51,52]. As will be presented in this review, natural metal sulfides in mineralogy databases show marked variation in chemical composition, crystal structure and metal/sulfur valence states, and therefore have the potential for a wide variety of catalytic functions. However, prebiotic experiments conducted to date have rarely taken into account these diversities, and have typically only used a limited number of single-metal sulfides, such as FeS and NiS, with a few exceptions [14,15,53], most likely due to the lack of the general principle to predict the catalytic function of natural mineral samples. Notably, in the field of heterogeneous catalysis, earth-abundant transition metal sulfides have attracted keen attention as cost–effective catalysts for energy conversion applications [54]. In particular, the hydrogen evolution, hydrogen oxidation, CO_2_ reduction, nitrate/nitrite reduction, and N_2_ reduction catalyzed by metals, metal oxides, and metal sulfides have been analyzed to identify the structural and electronic features of intermediates involved in the targeted reactions and determine the reaction pathways. Moreover, the application of the advanced theory of proton-coupled electron transfer allows prediction of how the activity and selectivity of prebiotic catalysts are potentially influenced by solution pH and Eh [55,56,57].

In the present review, we highlight the significance of the chemical diversity and complexity of natural metal sulfides for facilitating prebiotic chemical evolution and propose rational approaches for screening mineral catalysts from mineralogy databases. First, prebiotic synthesis experiments conducted to date using sulfides as catalysts will be summarized and the chemical diversity of metal sulfides in the mineralogy database will be surveyed with emphasis on Cu, Co, Fe, Mn, Mo, Ni, V, and W as the metal components, as these elements contain multiple accessible oxidation states and are involved in biological redox reactions with sulfur as a pervasive ligand. Transition metal sulfides have been proposed to have an evolutionary significance as protoenzyme catalysts [4,16,58,59]. Further, the structure-function relationship of metal sulfides is discussed based on material and catalysis engineering research findings. In addition, studies on metal sulfides with well-defined crystal phases, chemical composition, and defect structures are reviewed. Finally, theoretical and experimental approaches for probing structure-function relationships (i.e., how the structure of active sites dictates the overall performance of catalysts) and predicting heterogeneous catalytic properties are introduced. By presenting this integrated information, we hope to promote the screening and performance prediction of sulfide catalysts in prebiotic processes that preceded the origin of life.

## 2. Catalysis on Metal Sulfides Relevant to Prebiotic Chemistry

### 2.1. Thermal Catalysis on Metal Sulfides

Experiments on sulfide-promoted prebiotic reactions reported to date can be divided into two categories: non-catalytic (stoichiometric) and catalytic. A representative example of a non-catalytic process is associated with Wächtershäuser’s proposal of an “iron–sulfur world”, where FeS oxidation from mackinawite to pyrite (FeS + H_2_S → FeS_2_ + 2H^+^ + 2e^−^) provides reducing energy for organic syntheses. Huber and Wächtershäuser demonstrated that the reductive amination of α-keto acids to the corresponding α-amino acids (e.g., pyruvate + NH_3_ + 2H^+^ + 2e^−^ → alanine + H_2_O) occurs on freshly precipitated FeS, which was simultaneously oxidized to Fe^3+^ [50]. Interestingly, the reaction showed a strong metal specificity, as replacing FeS with Ag_2_S, CoS, Cr_2_S_3_, CuS, MgS, MnS, NiS, or ZnS did not result in amino acid formation. Heinen and Lauwers showed that CO_2_ was reduced to alkane thiols in a serum bottle containing FeS, H_2_S, water, and pure CO_2_ gas or a mixture of N_2_ and CO_2_ gases [60], with the formation of pyrite and H_2_ as byproducts. 

In addition to carbon fixation, ammonia serves as an important nitrogen source for the synthesis of amino acids, nucleotides, and other N-containing biomolecules [1]. Therefore, prebiotic ammonia synthesis has been investigated by reducing either nitrate/nitrite anions or dinitrogen. FeS can reduce nitrate and nitrite to ammonia in an acidic solution (pH 4) with the simultaneous oxidation of FeS to Fe_2_O_3_ [46]. The reaction was inhibited effectively in the presence of either Cl^−^, SO_4_^2^^−^ or H_2_PO_4_^−^, likely owing to the low binding affinity of nitrate on the catalyst surface. Brandes et al. [61] reported that pyrite and pyrrhotite reduce nitrate and nitrite to ammonia at a high yield (21%~89%) at temperatures between 300 and 700 °C and pressures of 0.1–0.4 GPa. These researchers also observed that Fe/H_2_O/N_2_ or Fe_3_O_4_/HCOOH/N_2_ systems reduce dinitrogen to ammonia at optimal temperatures of 700 and 500 °C, respectively, and a pressure of 0.1 GPa, albeit with a lower yield than that obtained from nitrate/nitrite reduction [61]. Based on these results, an oceanic or hydrothermal source for reduced nitrogen on early Earth was proposed. Dörr et al. [62] demonstrated the reduction of N_2_ to ammonia on freshly precipitated FeS using H_2_S as a reductant under milder conditions (under 1 bar N_2_ at 70–80 °C) with a yield of 0.1%. The Fe-S cluster-like surface moieties, which resemble the active center of FeMoS nitrogenase, were proposed to play a key role in the N_2_ activation.

For catalytic conversions of carbon and nitrogen-bearing molecules, Huber and Wächtershäuser reported that FeS, NiS, and (Fe,Ni)S provide favorable surface sites for activated thioester (CH_3_-CO-SCH_3_) formation from CO and CH_3_SH [47] and for amino acid polymerization via CO and H_2_S condensation to COS [49]. The former reaction resembles that found in the reductive acetyl-coenzyme A pathway, in which Ni and Fe centers are responsible for binding CH_3_SH and CO, respectively. Cody et al. [63] examined the condensation of alkylthiol (RSH; R = nonyl group) and CO under high temperature (250 °C) and pressure (200 MPa) condition in the presence of FeS and detected a small amount of pyruvate formation. FeS was inferred to activate pyruvate synthesis by forming a carbonylated iron sulfur complex (Fe_2_(RS)_2_(CO)_6_). Further, Cody et al. [53] assayed several nickel, cobalt, iron, copper, and zinc containing sulfides for hydrocarboxylation under the same high temperature and pressure condition and found that carboxylic acid was synthesized via carbonyl insertion at metal-sulfide-bound alkyl groups. With the exception of CuS, all of the metal sulfides promoted hydrocarboxylation. Because many transition metal complexes, including tertiary phosphate complexes of Co^2+^, Fe^2+^, Mo, and Ni^2+^, CH_3_Mn(CO)_5_ [64,65,66], are capable of mediating carbonyl insertion, similar chemical reactions can potentially proceed on various types of metal sulfides. In presence of FeS minerals (pyrrhotite, troilite, pyrite, and arseno-pyrite) together with H_2_, H_2_S, and CO_2_, Novikov et al. [67] demonstrated the conversion of pyruvate to lactate and/or reduced aldol compounds at more moderate temperatures (25–110 °C). These reactions were considered to be important because of their participation in extant metabolic pathways. Notably, addition of NH_4_Cl induced the reductive amination of pyruvate to alanine. Metal sulfides can also facilitate the condensation of formamide to purine and pyrimidine nucleic bases. Saladino et al. [68] conducted a heating experiment of formamide in the presence of FeS and (Fe,Cu)S minerals as catalysts at 160 °C for 48 h. In contrast to purine as the only product in the absence of any catalyst, several types of precursors of primary metabolism, such as purine and adenine, and of nucleobase isomers, such as isocytosine, 2-aminopurine, and 2(1*H*)-pyrimidinone, were synthesized. The presence of iron was a crucial factor for the synthesis of purine derivatives. Sulfides with both Fe and Cu, including chalcopyrite FeCuS_2_ and tetrahedrite (Fe,Cu,Sb)S, show enhanced yield towards isocytosine and 2(1*H*)-pyrimidinone products.

### 2.2. Electrocatalysis on Metal Sulfides

Recently, a model of geoelectrochemistry driven organic synthesis was proposed based on the detection of electrical current flow in natural deep-sea hydrothermal fields [13,17,18] and demonstration of the relevant organic syntheses in a laboratory setting [14,15,16,19,21,38]. The temperature, pH, and chemical composition differences between hydrothermal fluid and seawater generate a steep redox gradient across sulfide-rich vent rocks (Figure 1). This unique geoelectrochemical environment results in continuous and unidirectional electron flow from the internal hydrothermal fluid to the external chimney rock wall by coupling the oxidation of H_2_ or H_2_S in the hydrothermal fluid with the reduction of oxidative species in sea water, such as H_2_O, CO_2_, and nitrate/nitrite. The wide range of redox potentials and pHs in this environment control the kinetics and selectivity of target reactions even under chemically messy conditions, circumventing the lack of stereoselectivity seen in modern enzymatic systems [14,15,16]. As the reaction energetics and kinetics are dependent on multiple parameters (Eh, pH, and T), the distribution of products can be tuned based on the wide variation of geochemical environmental parameters, which have the potential to be rationally predicted based on the theory of proton-coupled electron transfer [55,56,57]. In addition, if the electrochemical fuel cell model (two-container model for prebiotic reactions; Figure 1) is applied to prebiotic reactions, even strong endothermic reactions can be initiated, as the pH (ΔpH) and temperature gradients (ΔT) across the electrically conductive, yet thermally insulative vent wall [68] can generate high-energy electrons at cold, ocean-vent interfaces [13,14,15,16,17,69].

Based on the electrochemical fuel cell model, Yamaguchi et al. [15] demonstrated the electrocatalytic reduction of CO_2_ to CO and CH_4_ on greigite (Fe_3_S_4_) under simulated alkaline hydrothermal vent conditions. As CO_2_ reduction will not proceed without overcoming the high energetic barrier associated with the first electron uptake step (‒1.9 V vs. the standard hydrogen electrode, SHE at pH 7) and also competes with the thermodynamically and kinetically more favorable reduction of protons to H_2_, the Faraday efficiency for CO production (CO_2_ + 2H^+^ + 2e^−^ → CO + H_2_O) is low (<0.02% in total at ‒1.3 V vs. SHE) at pH 5.5. However, by doping FeS with Ni to form FeNi_2_S_4_ (violarite), the CO_2_ electroreduction selectivity is enhanced by ~85-fold concurrently with a lowered onset potential (‒0.5 V vs. SHE, pH 5.5). CO_2_ reduction activity is further enhanced by modifying Ni-doped FeS catalyst with amine compounds [15]. Yamaguchi et al. have also found that the onset potential for CO_2_ reduction on Ni-doped FeS catalyst is ~180 mV more negative than the redox couple of H^+^/H_2_ at pH 5.5, suggesting that a naturally occurring proton motive force (PMF), which is equivalent to a 3–4 pH gradient, is sufficient to initiate CO_2_ reduction to CO when the hydrothermal fluid pH is higher than 9. Note that violarite resembles the NiFe_4_S_4_ active center of carbon monoxide dehydrogenase, and the amine species further enhance the CO_2_ reduction activity, similar to the peptide amine group surrounding the enzymatic active center (ligand-assisted catalysis). Amine-induced enhancement was associated with the ability of violarite to donate electrons to the anti-bonding π* orbital of CO_2_, which is essential for the efficient electrochemical activation of CO_2_ [69]. 

A study by Kitadai et al. [14] using various single-metal sulfides (Ag_2_S, CdS, CoS, CuS, FeS, MnS, MoS_2_, NiS, PbS, WS_2_, and ZnS) demonstrated that CdS and Ag_2_S function as excellent CO production catalysts (CO_2_ + 2H^+^ + 2e^‒^ → CO + H_2_O). The Faraday efficiency was as high as 40% on CdS at ≤−1.0 V (vs. SHE) at pH 5.5–6. The selective production of CO from CO_2_ by CdS and Ag_2_S is remarkable, because even though CO is frequently used as a carbon source for organic synthesis through thermocatalysis, CO is considered to have been present in only trace amounts in the Hadean atmosphere [70]. However, direct CO_2_ fixation into C_2_‒C_3_ compounds, such as acetate and pyruvate, has not been demonstrated in the presence of metal sulfides under electrochemical conditions. One exception was reported by Roldan et al. [38], who demonstrated CO_2_ reduction on highly faceted greigite nanoparticles into a series of C_1_‒C_3_ compounds, including formic acid, acetic acid, methanol, and pyruvic acid. In contrast to the reports by Yamaguchi [15] and Kitadai [14], Roldan et al. [38] cycled the electrode potential continuously during electrochemical CO_2_ reduction between 0.2 and −0.8 V vs. SHE (pH 6.5) at a slow rate (1 mV s^−1^), which may have promoted the formation of C‒C bonds.

In the geoelectricity driven prebiotic catalysis model, trace metal elements, such as Mo and W, could have played a significant role in prebiotic chemistry, as the continuous supply of electrons to the metal sites maintains the catalytic cycle. For example, Li et al. [16] demonstrated that molybdenum sulfide (MoS_2_) and Mo-doped greigite serve as electrocatalysts for reducing nitrate and nitrite to ammonia. It was found that in the presence of anions, such as phosphate, sulfate, iron sulfide impairs nitrate reduction ability, similar to the inhibition observed on FeS by Summers et al. [46]. In contrast, molybdenum sulfide can catalyze both nitrate and nitrite reduction over a wide pH range (pH 3–11) in the presence of phosphate and sulfate anions. In a neutral medium using molybdenum sulfide as the catalyst, ammonia was generated through nitrate reduction at an onset potential of 0 V vs. RHE (reversible hydrogen electrode). At more positive potential regions (0.6~0 V vs. RHE), not only ammonia, but also strong oxidants such as nitric oxide (NO) and nitrous oxide (N_2_O) were generated from nitrite and showed a potential-dependent distribution. This result suggests that the proton motive force is not required for ammonia synthesis. Notably, greigite doped with 5 atom % Mo efficiently activated nitrite reduction, suggesting that even at trace amounts, Mo can function as a catalytic site.

As a short summary, previous prebiotic chemical syntheses conducted using metal sulfides as the catalysts showed that in addition to iron sulfide, other transition metal sulfides can potentially function as C and N fixation catalysts with even higher activity than iron sulfides. The catalytic performance of CO_2_ reduction varies largely depending on the doping elements, crystal size, and surface structure of greigite, as well as the pH and Eh conditions [14,15,38]. Therefore, due to the chemical diversity of metal sulfides, in addition to the influence of pH and Eh on catalytic activity, suitable catalysts in mineralogy database with optimized activity can be potentially identified by screening reaction conditions. Effective stoichiometric C and N fixation requires an abundant source of electrons. In addition, Fe^2+^ was likely a dominant dissolved/colloidal metal species in the early ocean. Nevertheless, if geoelectrical currents had supplied energy to trigger prebiotic reactions, even trace amounts of metal elements could have continuously catalyzed the reaction in the geoelectrochemical environments in the Hadean ocean floor [16].

The above rationale and experimental evidence suggest that prebiotic metal sulfide catalysts should be screened not only based on the abundance of the metal, but also the electrocatalytic activity. In comparison to iron sulfides, the electrochemical and electrocatalytic properties of other transition metal sulfides and their mixtures remain poorly understood, with a limited number of studies conducted to date [14,15,53]. Proposed reaction mechanisms of these compounds are based on their resemblance with the relevant enzymatic processes, such as those mediated by carbon monoxide dehydrogenase [15], acetyl-CoA synthase [53,63], and nitrogenase [62], although no sufficient spectroscopic evidence and/or valence state information has been provided. The underlying reaction mechanisms should therefore be investigated in more detail for guiding catalyst screening based on functional, rather than structural relevance to well-characterized enzymes.

## 3. Chemical Diversity of Metal Sulfides in Mineralogy Libraries

As the coordination and electronic structure of mineral surfaces largely determines catalytic activity, here we evaluate the chemical diversity of naturally occurring metal sulfides to aid in the screening and identification of suitable prebiotic sulfide catalysts. A special emphasis was placed on the sulfides containing Co, Cu, Fe, Mn, Mo, Ni, V and/or W, as they are well-known redox-active elements and present in the active centers of enzymes catalyzing carbon and nitrogen conversions, such as those mediated by carbon monoxide dehydrogenase (Fe, Ni, Cu and Mo), a cobalt cobalamin cofactor for methyl transfer in acetyl-CoA synthase (Co), nitrogenase (Mo, V and Fe), nitrate reductase (Mo), nitrite reductase (Fe, Cu, and Mo), formate dehydrogenase (Mo and W), aldehyde oxidoreductase (Mo and W), and nitrous oxide reductase (Cu). In these enzymatic processes, transition metals switch oxidation states for charge accumulation, coordinate to a variety of substrates and intermediates, and also catalyze enzymatic redox reactions involving simple inorganic molecules, such as H_2_, AsO_2_^−^, S_x_^n‒^, as substrates, which function as coupling reactions for energy conversion and conservation in organisms. Sulfur is a pervasive ligand in the first coordination sphere of the enzymatic cofactors, although cobalt cobalamin is an exception, as tetrapyrrole serves as the ligand. The ubiquitous role of metals and sulfur in metabolic processes has led to the hypothesis that metal sulfides worked as “proto-enzymes” for prebiotic organic syntheses, and sustained the continuous chemical evolution toward functioning in complex biochemical systems [4,16,58,59].

Among the 5327 mineral species discovered to date (as officially listed by the International Mineralogical Association (IMA); http://rruff.info/ima/, as of 18 May 2018), 304 metal sulfides contain Co, Cu, Fe, Mn, Mo, Ni, V and/or W with 135,434 species-locality pairs. We evaluated the diversity and distribution of these metal sulfides based on the following criteria:(1)Locality frequency. This parameter reflects the availability of minerals in natural environments. Although detection limits and metal-specific biases reflecting the metal’s economic value may exist, an estimate of the probability of natural occurrence is useful for evaluating the feasibility of mineral-promoted prebiotic reactions.(2)Multiple metal composition. Naturally observed metal sulfides are commonly coupled to more than one transition metal and occasionally exhibit ternary compositions. Because cation substitutions and the presence of dual metal surface sites influence catalytic activities of minerals, as described in Section 2.2, Section 4 and Section 5, metal-specific behaviors and tendency to form multiple compositions are important factors for screening potential mineral catalysts.(3)Crystal structure and valence state of dominating elements. These two properties, particularly the valence state, control the electronic characteristics of the mineral. Valence state information is available for 89% of the localities of metal sulfides in the database (http://rruff.info/ima/).

### 3.1. Distribution of Transition Metal Sulfides

The locality count indicates the frequency that a certain mineral will be found within a wide range of geological localities. Although this parameter partially reflects the natural occurrence probability within the ability of human detection, it is useful for evaluating the relative importance of the minerals as geochemical catalysts. Among the 304 metal sulfides considered, Fe-containing sulfides dominate the library (Figure 2). Pyrite (Fe^2+^S^−^_2_) is the most frequently found species, followed by chalcopyrite (Cu^+^Fe^3+^S^2−^_2_), pyrrhotite (Fe^2+^_5_Fe^3+^_2_S^2−^_8_), arsenopyrite (Fe^2+^(AsS)^2−^), molybdenite (Mo^4+^S^2−^_2_), chalcocite (Cu^+^_2_S^2−^), marcasite (Fe^2+^S^−^_2_), bornite (Cu^+^_5_Fe^3+^S^2−^_4_), and tetrahedrite (Cu^+^_10_Cu^2+^_2_Sb^3+^_4_S^2−^_13_). The locality counts of these minerals exceed 5000 and together, these minerals comprise ~80% of the sulfide minerals discovered to date. The nine most frequently observed minerals may have been present since the middle Archean eon (~3500 Ma) or perhaps even earlier, although directly dated sulfides are rare.

Among transition metal sulfide species in the library, ~39% (119 species) contain iron, and are found in ~74% of localities (Appendix A), suggesting that iron was a pervasively occurring element during sulfide mineral formation. Co, Mn, Mo, W and V are mostly seen in Fe-free sulfides (>93% relative to the respective sulfides’ locality counts), whereas Cu and Ni show a high tendency to coexist with Fe. Cu-Fe- and Ni-Fe-containing sulfides are seen in 59% and 45% of localities of Cu- and Ni-sulfides, respectively, and exhibit structural variations accounting for 32% and 42% of the total number of Cu- and Ni-sulfide species, respectively. Nevertheless, Fe-free Cu and Ni single-sulfides exhibit high locality counts (24,085 and 2848, respectively), among which chalcocite (Cu^+^_2_S^2−^) and millerite (Ni^2+^S^2−^) are particularly prevalent (5520 and 1007 localities, respectively). Thus, although Cu and Ni tend to easily incorporate into Fe-bearing sulfides, their Fe-free sulfide forms are also ubiquitous. Please refer to Appendix A for more information regarding the spatial distribution and compositional diversity of metal sulfides.

### 3.2. Single-Metal Sulfides

Single-metal sulfide minerals have been mainly investigated as key prebiotic catalysts [14,16,38,46,47,48,49,50]. By studying this family of minerals, their intrinsic metal-specific properties can be derived and compared, as was shown by Kitadai et al. [14] through the intensive screening of catalysts for CO_2_ fixation. Single-metal sulfides of Fe, Cu, and Ni show marked variations in crystal structure, chemical composition and metal/sulfur valence states. Among 45 Fe single-metal sulfides (Figure 3), two pairs of polymorphs are present with the chemical compositions of FeS_2_ (pyrite and marcasite) and FeS (troilite and keilite). Pyrite and marcasite have indirect band gaps of 0.7 and 0.4 eV, respectively [71]. Orthorhombic marcasite exhibits an anisotropic property that results in a unique sulfur surface site (sulfur trimers) on a specific surface orientation [72]. In contrast to pyrite on which S^2−^ is firstly oxidized, the sulfur trimer species on marcasite are most easily oxidized.

Fe single-metal sulfides in 91% of localities have valence state information. Sulfides having Fe^2+^ alone dominate the mineral family (84.86%), followed by those containing both Fe^2+^ and Fe^3+^ (15.12%), and by three sulfides that only contain Fe^3+^ (0.02%). Sulfides composed of a mixture of Fe^2+^ and Fe^3+^, such as pyrrhotite (Fe^2+^_5_Fe^3+^_2_S^2−^_8_) and greigite (Fe^2+^Fe^3+^_2_S^2−^_4_), typically exhibit magnetic properties [73,74,75].

Cu-, Co-, and Ni-single sulfides also display diverse chemical and physical properties (Appendix A). As many as 119 species were identified in Cu single-metal sulfides, of which 27 species have valence state information (73% relative to the locality counts of Cu single-metal sulfides). In addition, 18 species contain only Cu^+^ (59%), four (tetrahedrite, tennantite, famatinite, and kuramite) are formed from Cu^+^ and Cu^2+^ (40%), and the remaining 5 species possess only Cu^2+^ (<0.6%). The percentages in parentheses represent the respective mineral distributions relative to the locality counts of Cu single-metal sulfides with valence state information.

Co single-metal sulfides form 10 different structures, including five Co^3+^-sulfides with two sulfosalt polymorphs, Co^3+^(AsS)^3−^ (cobaltite and alloclasite) and Co^3+^(SbS)^3−^ (willyamite, costibite, and paracostibite). Linnaeite (Co^2+^Co^3+^_2_S_4_) is the only species having a mixed valence state. The environmental distributions of Co^3+^-, Co^3+^ plus Co^2+^-, and Co^2+^-sulfides are 78.93%, 18.47%, and 2.59%, respectively, relative to the total locality counts of Co single-metal sulfides.

For Ni single-metal sulfides, the valence state of Ni is always +2, with the exception of polydymite, which has both Ni^2+^ and Ni^3+^ (Ni^2+^Ni^3+^_2_S^2−^_4_).

In Mn, Mo, and W single-metal sulfides, the valence states of metals are conserved, with only Mn^2+^, Mo^4+^, and W^4+^ being observed.

For V single-metal sulfides, three species are known (patrónite (VS_4_), colimaite (K_3_VS_4_), and yushkinite (Mg,Al)(OH)_2_VS_2_), in which the valence state of vanadium is +4 or +5. The occurrence of these species in natural environments is rare, and V-bearing sulfides are typically found as Cu–V binary forms (95% in terms of locality).

Generally, metals with low valence states predominate the library of metal sulfides, with the exception of Co, allowing these minerals to act as an electron source or a catalytic center for charge accumulation during redox conversion. In addition, sulfur with a −1 valence state, which typically exists as a disulfur (S_2_)^2−^ ligand, can act as an electron-accepting site, and thus has the potential to be involved in redox catalysis. The participation of not only metal sites, but also sulfur ligands, during redox catalysis has been extensively reported and will be discussed in Section 4.3.

### 3.3. Binary- and Ternary-Metal Sulfides

As described in Section 2.2, the doping of a secondary metal in iron sulfide can enhance the activity and selectivity of CO_2_ reduction [15]. Sulfides with binary metal composition can thus potentially serve as superior catalysts to their single-metal counterparts. In addition, as binary metal sites are present in several extant enzymatic systems, such as Ni-S-Fe hydrogenase, Ni-S-Fe carbon monoxide dehydrogenase (CODH), Mo-S-Cu CODH, Fe-S-Mo nitrogenase, and Fe-S-V nitrogenase, it has been suggested that binary metal centers can synergistically function as Lewis acid and base sites for facilitating catalysis. For example, Fe and Ni bind CO_2_ in a bidentate mode via O and C atoms, respectively [78].

Cu-bearing binary sulfides exhibit 193 structures among 58,932 localities and include 52 species of Cu-Fe binary sulfides (59% in terms of locality) (Appendix A). Calcopyrite (Cu^1+^Fe^3+^S^2−^_2_) and bornite (Cu^1+^_5_Fe^3+^S^2−^_4_) are the most prevalent forms in this category. Sulfides with Fe and Cu as substitutional cations are also known, but are rare in terms of distribution (<1% of localities).

Ni-bearing binary sulfides with 38 structures have been identified among 5185 localities. Of these species, 10 are Ni-Fe binary sulfides (45% in terms of locality) (Appendix A), of which 7 contain Ni and Fe as substitutional cations with an Ni/Fe ratio of 0–35 atomic % being reported [79]. These seven sulfides cover 75% of localities of Ni-Fe binary sulfides with pentlandite (Ni,Fe)_9_S_8_ being the most prevalent form. The remaining 28 species possess Fe and Ni with fixed stoichiometries (e.g., violarite (FeNi_2_S_4_)), in which Fe and Ni occupy independent crystal lattice sites. The capability of Ni-Fe binary sulfides to have both fixed and varied Ni/Fe ratios in their structures is a unique characteristic different from that of Cu-Fe binary sulfides, in which Cu and Fe tend to form specific structures with fixed stoichiometries.

A number of ternary-metal sulfides with combinations of Fe–Cu–Ni (seven species), Fe–Cu–Mo (two species; tarkianite ((Cu,Fe)(Re,Mo)_4_S_8_) and maikainite (Cu^+^_10_Fe^2+^_3_Mo^4+^Ge^4+^_3_S^2−^_16_)), and Fe–Cu–W (one species; ovamboite, Cu^+^_10_Fe^2+^_3_W^4+^Ge^4+^_3_S^2−^_16_) have been characterized (Appendix A). The occurrence of ternary systems is uncommon (77 localities or 0.05% of the total locality counts of metal sulfides) and it is notable that binary Fe-Mo and Fe-W sulfides are not found in nature, indicating that Cu is important for the incorporation of Mo and W into FeS structures.

### 3.4. X-ray Amorphous Metal Sulfides

As a clear XRD pattern is required to identify crystal structures, X-ray amorphous species are rarely included in mineralogy databases (Appendix A lists the eight amorphous species recorded in the RRUFF database). Jordisite (MoS_2_), the second most prevalent Mo single-metal sulfide, is the only amorphous metal sulfide recorded. Using high-resolution transmittance electron microscopy (TEM), jordisite exhibits a low crystallinity, but still shows a layered structure with interlayer spacing of ≈6 Å resembling that in hexagonal molybdenite (MoS_2_) [32]. Mo sulfide is also seen in anaerobic sulfidic sediments and basins [28,30,80,81,82], and its speciation is controlled by the O_2_ and HS^−^ concentrations in these environments [28,29]. Mo sulfide associated with Cretaceous deep-sea sediments and with other black shales is characterized by the molecular structure of the Mo center coordinated with sulfur and oxygen, as resolved by extended X-ray absorption fine structure spectroscopy [30].

Despite of the low number of amorphous species in the database, metal sulfide species with a low crystallinity are not rare. Aqueous clusters of FeS, ZnS and CuS constitute a major fraction of the dissolved metal load in anoxic oceanic, sedimentary, freshwater and deep ocean-vent environments, and may be possible building blocks during mineral formation [83]. Luther et al. [83] reported that Fe-S, Zn-S, Ag-S clusters with sizes <3 nm form in the presence of low concentrations (μM order) of metallic ions and sulfides and show remarkable kinetic stability. For example, the half-life of as-formed Ag-S clusters with respect to O_2_ oxidation is 360 days. In addition, these nanoclusters show a wider band gap in comparison to their bulk counterparts owing to the quantum confinement effect. Cu-S clusters also show wide variation in metal/sulfur stoichiometry (neutral Cu_3_S_3_ and anionic Cu_4_S_6_ clusters). Nanoclusters with varied metal/sulfur stoichiometry are either neutral or negatively charged and are soluble in aqueous solutions. With the advance of atomic resolution structural characterization techniques, such as X-ray absorption spectroscopy and transmittance electron microscopy, low-crystallinity metal sulfide species are expected to be more frequently discovered in geochemical environments. As such amorphous materials adopt a metastable state and have numerous “dangling bonds”, the unique band structure and surface properties of X-ray amorphous metal sulfides are expected to have unique catalytic properties in comparison to that of their bulk counterparts.

## 4. Impact of Metal Sulfide Chemical Diversity on Catalysis

The previous section summarized the marked chemical diversities of natural transition metal sulfides, as exhibited by the large variations in chemical composition, crystal structure and valence states of the component metal and sulfur species. The chemical diversity of metal sulfide minerals partially reflects the “messy” feature of geological environments. In the context of prebiotic chemistry, such diversity among the available catalysts could have generated messy chemistry with an enormous number of possible reactions, leading to the formation of complex prebiotic organic synthesis networks and, ultimately, autocatalytic reaction networks [84,85]. This raises the question of what types of sulfides, or their combinations, could have been involved in prebiotic processes toward the origin of life?

To extract valuable information from mineral databases for the screening of prebiotic catalysts, a critical guiding principle is the “structure–function relationship”, including the key reaction intermediate(s) and physicochemical factor(s) dictating catalytic activity. As was introduced in Section 1 and Section 2.2, electrocatalytic processes can occur in hydrothermal vent environments for driving prebiotic synthesis. More importantly, the experimental methods and theories developed in the electrocatalysis field can be generally applied for all redox and a portion of non-redox catalytic processes. For instance, redox processes occurring in enzymes have been extensively studied using electrochemical methods [86,87,88], thereby allowing detailed elucidation of the electron and proton- transfer mechanisms. To provide a better understanding of structure–function relationships toward rational catalyst screening, here, we review the electrocatalytic properties of sulfides with well-defined structures that have been extensively studied in engineering fields.

### 4.1. CO_2_ Reduction

CO_2_ electroreduction has been examined on various types of catalysts, including metals, alloys, metal–organic complexes, homogeneous molecular catalysts, and metal sulfides, in both aqueous solutions and organic solvents [89,90,91,92]. The products of these reactions are typically simple C1 compounds, such as CO and formate, in water or C2 compounds, such as oxalate, in aprotic solvents [89,90], although Cu was reported to efficiently catalyze CO_2_ reduction to hydrocarbons, including methane and ethylene, in water [93,94].

Metal sulfides generally exhibit lower catalytic activity for CO_2_ reduction than their metal counterparts [14,15,94], but the presence of favorable active sites on metal sulfides is computationally predicted to facilitate the CO_2_ reduction by stabilizing the key reaction intermediates [95]. The edge sites of MoS_2_ and Ni-doped MoS_2_ preferentially bind COOH* and CHO* to the bridging S atoms and CO* to the metal sites. The binding of intermediates on different sites is expected to decrease the overpotential owing to the intrinsic thermodynamic adsorption energy scaling relations between different intermediates [96,97]. In fact, selective CO_2_ electroreduction to CO has been reported on MoS_2_ in water containing 4 mol% of an ionic liquid (1-ethyl-3-methylimidazolium tetrafulorobrate; EMIM-BF_4_) [98]. CO production occurred at an onset potential as low as 0.054 V (vs. RHE) and the Faraday efficiency increased from <10% to close to 100% by increasing the overpotential from 0.2 to 0.76 V. Notably, no appreciable amount of CO was observed in the absence of ionic liquid. Density functional theory (DFT) calculations suggested that the *d*-orbital electrons of Mo dominating near the Fermi energy level are freely transferred to reactants adsorbed on edge Mo sites. EMIM ions were inferred to stabilize CO_2_ via hydrogen bonding to form an [EMIM-CO_2_]^+^ complex and inhibit the conversion of CO_2_ to HCO_3_^−^ and CO_3_^2−^. The complexation also lowers the energy barrier for electron transfer from MoS_2_ to the physisorbed CO_2_ on MoS_2_.

Besides the artificial catalytic systems, the mechanistic implications of CO_2_ reduction can be obtained from analyzing the catalytic process in biological enzymatic systems. The reversible transformation of CO_2_ to CO is catalyzed by a highly asymmetric [NiFe_4_S_4_] cluster in anaerobic carbon monoxide dehydrogenase (CODH), which operates at potentials close to the thermodynamic potential of the CO_2_/CO couple (−0.52 V vs. SHE, pH 7) and contains bifunctional metal sites. In Ni-Fe-containing CODH, high-resolution crystallographic structure determination [78] revealed that CO_2_ undergoes two-electron reduction after binding to Ni and Fe sites with a μ_2_, η^2^ mode, and the intermediate with bent-bound CO_2_^2−^ is stabilized by substantial π backbonding. The bifunctionality of Ni and Fe to serve as nucleophilic and electrophilic sites, respectively, plays a key role in activating CO_2_ and promoting electron transfer from the metal to CO_2_. A short length of Ni-C bonds suggests that the Ni center has high nucleophilicity, which likely originates from the integration of Ni into the Fe/S scaffold. Based on these enzymatic properties and mechanisms, sulfide minerals bearing Ni and Fe, which have an Ni site of square planar symmetry and a single unsaturated site, could potentially act as CO_2_ reduction catalysts. 

### 4.2. Ammonia Synthesis from Nitrate/Nitrite or Dinitrogen Reduction

Electrochemical ammonia synthesis by nitrate/nitrite reduction was recently studied on synthesized molybdenum and iron sulfides [16,99]. Nitrate and nitrite reduction catalyzed by molybdenum sulfide have an evolutionary relevance with respect to enzymatic catalysis, because all nitrate reductase enzymes and one family of nitrite reductases rely on molybdenum-pterin (Mo-S_2_) active centers, in which Mo is coordinated by sulfur atoms of a pterin group [100]. He et al. [99] found that product selectivity of nitrite reduction on molybdenum sulfide is pH dependent with respect to the formation of NO, N_2_O and NH_4_^+^. Operando electron paramagnetic resonance (EPR) and Raman spectroscopies during nitrite reduction revealed that an oxo-ligated Mo(V) species serves as the key intermediate controlling the nitrite reduction selectivity [99]. Oxo-ligated Mo(V) species changes its configuration from an isotropic to distorted geometry in response to pH change, suggesting that these species decouple electron transfer (ET) from proton transfer (PT). This unique property results in a volcano-type pH-dependent selectivity of N_2_O formation, where the optimal pH for this reaction is close to the p*K*a of Mo^V^-oxo species, at which a kinetic balance is anticipated to exist between ET and PT.

Dinitrogen reduction is another candidate pathway to generate ammonia (N_2_ + 6e^−^ + 8H^+^ → 2NH_4_^+^), and several electrocatalytic systems using Fe_2_O_3_/CNT [101], Fe/CNT [102], and Ru/C [103], among others, as catalysts have been reported to function in aqueous solution at room temperature and ambient pressure. Typically, low Faradaic efficiency (<1%) was exhibited and H_2_ was generated as the main byproduct. The reaction rate was enhanced by increasing the temperature or optimizing the potential. Higher Faradaic efficiencies (4.02–10.1%) were recently reported by engineering the catalysts [104,105]. For example, through amorphourization of an Au nanocatalyst, structural distortion was observed, which was proposed to increase the concentration of active sites in the catalyst with higher binding affinities towards N_2_. To our knowledge, no metal sulfide has been reported to perform electrocatalytic N_2_ reduction. As FeMoS cofactor of nitrogenase catalyzes N_2_ reduction under ambient temperature and pressure, several Mo-containing molecular [106,107] and heterogeneous (MoS_2_, MoO_3_, MoN) [108,109,110] catalysts have been synthesized for N_2_ reduction. For example, Yandulov et al. [106] reported that a Mo triamidoamine complex catalyzes N_2_ reduction using decamethyl chromocene and pyridinium salts as electron and proton sources, respectively, at ambient temperature and 1 bar N_2_ in an aprotic organic medium. Based on the chemical properties of the isolated intermediates, the reaction is proposed to proceed via a distal associative mechanism, wherein the catalytic intermediates are sequentially protonated and reduced. A single molybdenum center is proposed to function as the active site and cycle from Mo(III) to Mo(VI) states. Note that the displacement of ammonia by N_2_ is the rate-determining step on this molecular catalyst, which differs from heterogeneous Au [104] and MoS_2_ [109] catalysts, where the reductive protonation of bound N_2_* is considered to be the rate limiting step based on DFT calculations to weaken the N≡N triple bond.

### 4.3. Hydrogen Evolution and Oxidation Reactions

The H_2_ evolution reaction (HER; 2H^+^ + 2e^−^ → H_2_ or 2H_2_O + 2e^−^ → H_2_ + 2OH^−^) has been the most intensively studied electrochemical reaction catalyzed by metal sulfides. As mentioned above, H_2_ evolution tends to compete with the reduction of N_2_, CO_2_ and NO_3_^−^ in regions of large overpotential. For this reason, understanding the HER mechanism is expected to aid in the rational design and selection of catalysts and conditions for the selective N_2_, CO_2_ and NO_3_^−^ reduction through suppression of HER. HER has been examined on various sulfides, including Fe [111], Co [112], Mo [113,114,115,116,117,118], and W [119,120] single-metal sulfides, Ni/Fe [121] and Co/Fe [122] binary-metal sulfides, Co-based binary-metal sulfides (with Cu, Ni, Zn) [123], and pyrite-type minerals [124]. Mo sulfide possesses a laminated lattice structure that is highly amenable to electric and structural modifications, and its catalytic property has therefore been investigated in different crystal phases, including hexagonal MoS_2_ (2H-MoS_2_), chemically exfoliated metallic MoS_2_ (1T MoS_2_), amorphous MoS_x_, Mo-S clusters (Mo_3_S_4_^−^), and Mo-Cu binary-metal sulfide (Cu_2_MoS_4_) [113,114,115,116,117,118]), and with various degrees of non-stoichiometric defects [115,117], doping [125,126], and hybridization [127,128,129]. From the results of spectroscopic analyses, including in situ and ex situ X-ray absorption [130], X-ray photoelectron spectroscopy (XPS) [131], EPR [132] and Raman spectroscopy [113,132], hexagonal MoS_2_ nanoparticles appear to have catalytic activities on edge sites, rather than on basal planes [114,115,132,133,134,135]. This possibility has been supported by several experimental studies that have used [Mo-S_2_]-bearing compounds, such as [Mo_3_S_13_]^2−^ and [(PY5Me_2_)MoS_2_]^2+^, which mimic edge sites [118,135,136,137]. 

Experimental evidence also suggests that terminal di-sulfur (S-S)_tr_ species are critical for the HER on amorphous MoS_x_ [113,130,131]. (S-S)_tr_ species provide thermodynamically favorable adsorption sites for hydrogen (∆_r_G^o^ ~0) by forming unsaturated –S sites through the reductive cleavage of the S-S bond [131,134]. The formation of a molybdenum hydride moiety (Mo-H) also triggers the HER on amorphous MoS_x_ [132] and on Mo_3_S_4_^−^ [138]. Recently, operando Raman spectroscopy revealed a dynamic structural change of a Mo tri-nuclear cluster characterized by Mo-Mo bond weakening and the concurrent emergence of a terminal S-S ligand, indicating that both multinuclear Mo-Mo bonds and terminal S-S ligands play synergistic roles in facilitating the HER [139]. A similar mechanism involving exposed disulfur ligands as active sites was proposed to explain the efficient HER that proceeds on amorphous CoS_x_ [112]. In both WS_2_ and MoS_2_, the 2H phase material converts into a metallic 1T phase when chemically exfoliated to monolayer thickness. The metallic 1T phase exhibits superior HER activity because of favorable electron transfer rate as well as the thermoneutral adsorption of H [119]. Based on these mechanistic studies, sulfides with S_2_^2−^ moieties, such as pyrite (FeS_2_), marcasite (FeS_2_), covellite (Cu^+4^Cu^2+^_2_(S_2_)_2_S_2_), vaesite (NiS_2_), and cattierite (CoS_2_), are promising catalysts for H_2_ evolution owing to the redox activity and suitable adsorption energy of the of S-S species towards H_ads_. 

The reverse reaction of the HER, namely the hydrogen oxidation reaction (HOR), is an important half reaction that is predicted to have occurred at the inner chimney wall–hydrothermal fluid interface of geoelectrochemical reactors, and to have supplied electrons for driving reductive organic reactions at the mineral–seawater interface (Figure 1). The HOR is catalyzed efficiently by Pt-group metals, such as Pt, Pd, Ir [140,141] in acidic medium. Moderate HOR activities have also been shown using Ni-based catalysts, such as Ni [142,143], Ni binary alloys (NiMo, NiTi) [144], and ternary metallic CoNiMo catalyst [145] in alkaline medium, and by Ni complex molecules [146] in organic medium. Based on DFT calculations, the H adsorption energy on the surface dictates the HOR activity. Pt shows thermoneutral adsorption, whereas Ni displays relatively higher adsorption energy (stronger binding). By alloying Ni with other metals, the H binding energy was tuned owing to the changes of the d-orbitals of Ni, which facilitates the HOR [144,145]. To date, no metal sulfide has been reported to catalyze the HOR, although numerous types of metal sulfides efficiently catalyze the HER, as mentioned above. This contrasts greatly with Fe-S-Ni hydrogenase, which reversibly and efficiently catalyzes both the HER and the HOR [88], and is also predicted to thermoneutrally adsorb H based on DFT calculations [134]. Based on Koper’s prediction [56], the presence of only one intermediate (generated through an electrochemical step) in a 2e^−^/2H^+^ reaction will allow the catalyst to effectively perform both the forward and backward reactions, given that thermoneutral adsorption of the intermediate occurs on the catalyst surface. One possible explanation for the inertness of metal sulfide towards the HOR is that another non-electrochemical step, such as the surface binding of H_2_, non-redox dissociation of H-H bond, or deactivation of the catalyst surface, is rate limiting. Because of the importance of the hydrogen oxidation reaction for geoelectrochemical organic syntheses, the HOR catalyzed by metal sulfides warrants further investigation.

The electrocatalytic studies reviewed above demonstrate that metal sulfides have the potential to function as versatile catalysts for redox reactions that are relevant to prebiotic synthesis. Several key factors of metal sulfide catalysts can be summarized here. First, in situ spectroscopic analyses have suggested that not only single-metal sites, but also heterogeneous surface moieties, such as those with metal-metal, metal-sulfur, metal-hydride, and sulfur-sulfur bonds, play important and occasionally synergistic roles in promoting specific reactions. This is a remarkable feature of metal sulfide catalysts that differs from that observed in metal complex-mediated reaction systems, in which the redox chemistry is mainly dominated by the valence change on single-metal centers. Moreover, a surface dominated by metal sites is likely preferable for CO_2_ and nitrate electroreduction on metal sulfides, as (di-)sulfur-dominated surface sites will be active for H_2_ evolution. Furthermore, the doping of a secondary metal was theoretically predicted to tune the binding energies of the reaction intermediates, such as CHO*, COOH*, and H*, thereby enhancing the CO_2_ reduction selectivity by suppressing competing H_2_ evolution [95]. Therefore, the binary- and ternary-metal sulfides in the mineralogy database, which includes 193 Cu-bearing binary-metal sulfides species, 38 Ni-bearing binary-metal sulfide species, and 10 types of ternary-metal sulfides, are anticipated to show unique activities in comparison with their single-metal sulfide counterparts. The microscopic mechanism of decoupled electron and proton transfer represents a new mechanistic aspect to mutli-electron, multi-proton reactions [55,56,57] and provides a rational approach to tuning catalytic reactivity and selectivity over wide pH and Eh ranges.

## 5. Rational Screening of Catalysts

The examples shown in Section 4 highlight the physicochemical factors influencing the catalytic activity of specific types of metal sulfide catalysts. To establish a complex chemical network composed of life’s building blocks and their polymers, significant efforts are required to screen and identify efficient mineral catalysts for activating simple molecules (CO_2_, N_2_, NO_3_^−^/NO_2_^−^, H_2_, etc.). Although a few prebiotic studies have attempted to screen metal sulfide catalysts [14,15,53], approaches for rational screening have not yet been described. In this section, we briefly review an in silico approach, followed by machine-learning methods (so-called catalytic informatics) for predicting the catalytic activity of electrocatalysts. 

Electrocatalysis occurs via the adsorption of substrates on active surface site(s), followed by the sequential conversion of the substrates into bound intermediates, and finally, the desorption of products from the catalytic surface. Efficient catalysts are expected to have adsorption energy towards the reactants and intermediates that are neither too high nor too low, because too weak binding will not drive effective conversion, whereas too strong binding will lead to the poisoning of active sites (the so-called Sabatier principle, d-band theory). Recently, the prediction based on the Sabatier principle has become a more powerful approach owing to the pioneering research by Nørskov et al. [96,125,147]. Instead of experimentally identifying the binding energy of a key intermediate, they developed the DFT methods to calculate the adsorption energy. This approach has been widely applied to rationalize the experimentally observed tendency of CO_2_ electroreduction reactivity on various metal catalysts and even predict new catalysts with better performance. For example, Figure 4a–c illustrates typical reaction pathways towards various products, particularly CH_4_, CO, HCOO^−^, CH_3_OH, and CH_3_CH_2_OH, as was summarized by Kortlever et al. [91]. Peterson and Nørskov [96] calculated the CO_2_-to-CH_4_ reduction pathway on a series of fcc transition metal (211) surfaces and showed that Cu locates at the top of the activity plot and is hence the best catalyst (Figure 4d). On Cu (211), the energetic barrier between the CO_2_ → COOH* conversion and that of the CO*→CHO* step was well balanced. The CO*→CHO* process is typically the potential-determining step and leads to a high overpotential (on the order of 1 V). The CO_2_ reduction pathways towards the products CO, C_2_H_4_, and CH_3_CH_2_OH were also calculated on Cu (100) and Cu (211) [148,149]. Generally, negatively charged intermediates, such as the adsorbed CO_2_^−^ anion intermediate and adsorbed (CO)_2_^−^ anionic dimer, are critical for C1 and C2 compound formation. 

Certain alloys facilitate redox reactions with greater efficiencies than the single-metal counterparts, as has been demonstrated for CO oxidation on Pt-alloy, methanation on Fe-Ni alloy, and ethylene oxidation on Ag-alloy [147]. Particularly, CO_2_ electroreduction on Cu-Ag alloy shows a higher selectivity for hydrocarbon and oxygenate (ethylene, ethanol, and propanol) formation in comparison with that on Cu [150]. He et al. [151] screened several binary- and ternary-metal alloys (In_x_M_1−x_; M = Fe, Co, Ni, Cu, Zn) for CO_2_ reduction to CO and found that In_x_Cu_1−x_ alloys with x = 0.2–0.5 exhibited the highest activity for the CO production (90% selectivity). The CO evolution on In_x_M_1−x_ occurred with the trend Fe < Co < Zn < Ni < Cu, whereas that for the H_2_ evolution followed the trend Zn < Cu < Ni < Co < Fe. The activity trend for the CO formation was consistent with the sequence of CO adsorption energy on the catalyst surfaces calculated by DFT, thus verifying the validity of CO adsorption energy as the key descriptor of catalytic activity. In these alloys, the secondary metal changes the average energy of the *d*-electrons (ε_d_–ε_F_; the *d*-band center with respect to the Fermi level) in the primary metal, resulting in either weakening or strengthening of key intermediate adsorptions. [147]. In addition to CO_2_ reduction, the catalytic activity for H_2_ oxidation [140,152,153] and N_2_ reduction [154,155,156,157] have also been examined by DFT calculations with d-band theory.

For reactions involving multiple intermediates, such as CO_2_ reduction to CH_4_ or alcohols, and N_2_ reduction to ammonia, the adsorption energy scaling relationship of chemisorbates commonly leads to intrinsic overpotentials [96,158]. It is possible to lower the onset potential of these reactions through the selective binding of intermediates on different sites or with different binding modes by decoupling the scaling relationship of the binding energies of intermediates [96]. By applying DFT calculations to a mineral with a known crystal structure and chemical composition, the electronic structure of the bulk catalyst and surface sites, including band gap and Fermi levels, can be derived for determining the adsorption energy of key intermediates, which can be used for predicting the activity and identifying superior catalysts (Figure 5). The Materials Project database [159] (https://materialsproject.org/) has been developed for this purpose and is supported by the U.S. Department of Energy (DOE). This database contains the crystal structures of 83,989 inorganic compounds (as of 31 July 2018). Using high-throughput computing, the band structures of 52,179 compounds have been calculated and can be used for calculating adsorption energy.

It is to be noted here that the aforementioned in silico approaches have now become a more exhaustive search for new catalysts owing to the integration to machine-learning methods. For example, Ma et al. developed a machine-learning-augmented chemisorption model that enables fast and accurate prediction of CO_2_ reduction activity of metal alloys [160] (Figure 6). With this model and the d-band theory, they identified promising new catalysts for selective reduction of CO_2_ to C2 species [160]. Ulissi et al. [161] developed a neural-network-based surrogate to share information between activity estimates with an order of magnitude fewer explicit DFT calculations. This approach allows for an order of magnitude reduction in the number of DFT calculations required and thus rapid prediction of catalysts even for highly complicated bimetallic electrocatalyst for CO_2_ reduction. Besides the use of in silico data for machine learning, a recent study by Shao-Horn and coworkers [162] showed the potential to utilize experimental data from various literature for catalyst prediction after standardizing with the activity of a material common in all studies. This approach can be considered as a technical breakthrough in this field, because one of the difficulties hindering catalyst informatics was the lack of data in a uniform structure. The implementation of an internal standard allows access to the wealth of experimental data accumulated to date, which may lead to a more widespread application of machine-learning techniques. More recently, Ooka et al. developed the method to utilize genetic sequence information for catalyst development [163]. In contrast to experimentally-determined activity which varies depending on the evaluation method (current vs. overpotential, or pH conditions), the unambiguity of the gene sequence allows for a more robust statistical analysis.

Owing to advances in protein crystallography, high-resolution enzyme structures are becoming increasingly available (e.g., 132,902 protein structures are deposited in the Protein Bank Database as of 9 August 2018). The Protein Bank Database provides structural information for numerous metal-sulfur enzymes, from which the metal-sulfur distance, sulfur-metal-sulfur angles, and coordination symmetry around the metal center can then be determined. The LUMO and HOMO energy levels of enzyme active centers can be calculated using quantum chemical methods based on simplified enzyme structural models [164,165]. These parameters can be compared with the corresponding crystal structures and band energies of inorganic minerals [71,166,167]. Such comparisons between the electronic structure of minerals and enzymes would aid the screening and identification of suitable mineral catalysts for specific target reactions.

Structure–function relationships can be reliably predicted by combining molecular-level information with quantum chemical calculations. As reviewed in Section 4, heterogeneous sulfide catalysts typically have surface states that are distinct from those of single-metal sulfides and can only be characterized by spectroscopic measurements. In situ spectroscopic monitoring of the structural evolution of catalysts provides molecular information of key reaction intermediates. The well-defined molecular structure of active species can further be applied in DFT calculations to derive the descriptor (adsorption energy of key intermediate) of the activity. This is particularly important in cases where the catalyst structure changes during charge accumulation and/or new intermediate adsorbates are formed. Amorphous materials are another example to be examined spectroscopically for elucidating reaction mechanisms [113,130,131].

Despite the recent advancement of catalytic screening mentioned above, several challenges remain to be made in order to effectively apply these approaches. Specifically, for DFT calculations, determining the descriptor in multi-step reactions is more difficult compared to single-step reactions, as the descriptor changes depending on the reaction conditions, such as the atmosphere and solvent composition, pressure, pH, and temperature. The amount of surface coverage by adsorbates and adsorbate–adsorbate interactions also adds to the complexity of such computations. In addition, during the reaction process, the original surface structure may be modified. Spectroscopic investigation of the intermediate processes is needed to trace these potential structural changes and provide an accurate molecular basis for DFT calculations.

## 6. Perspectives

Mineral-mediated catalysis is at the center of chemical evolution involved in the emergence of life. Interdisciplinary knowledge from the fields of catalysis engineering and mineralogy is needed to rationally explore catalytic processes that occur in plausible geochemical settings. The vast chemical diversity of metal sulfides is associated with marked variations in catalytic properties, which are influenced by metal composition, crystal structure, valence state, and defect structure. However, this diversity provides a possible solution to the current problem in prebiotic chemistry, where usually an inferior activity, poor selectivity of catalytic processes and low chemical diversity among the generated products were shown. As most studies have examined limited types of catalysts, considerable efforts are needed to examine the properties of various types of metal sulfides and their combinations, and to determine concrete structure–function relationships that can be used as a guide to search mineralogy databases for target catalysts. Modern computational chemistry, operando spectroscopy, and machine-learning approaches are critical for catalysis engineering by establishing the key “descriptor” of activity. Further advances in the understanding of electron and proton transfer and d-band theory, which is generally considered to be the most suitable approach for predicting electrocatalytic performance, will aid in the screening of mineral catalysts for prebiotic reactions crucial for the origin of life.

## Figures and Tables

**Figure 1 life-08-00046-f001:**
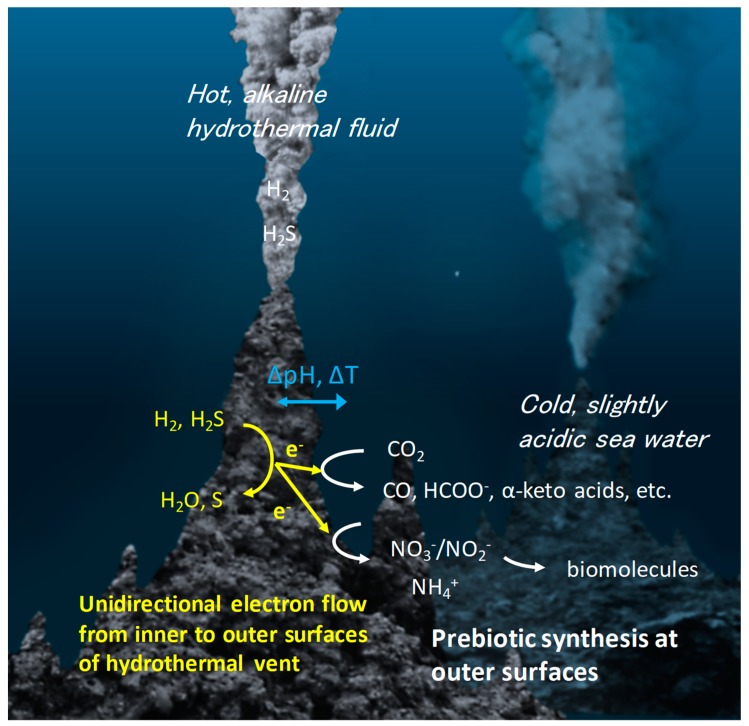
Model of geoelectrochemical prebiotic synthesis for chemical evolution. In deep-sea hydrothermal vents, electrons generated from the oxidation of H_2_ and H_2_S in the hydrothermal fluids flow towards the seawater-rock interface through the porous, conductive metal sulfide chimney minerals. The reduction of molecules, such as CO_2_, NO_3_^−^, NO_2_^−^, at the seawater-rock interface is catalyzed by metal sulfides to generate various types of life’s building blocks, which further undergo condensation or polymerization to bio-essential polymers.

**Figure 2 life-08-00046-f002:**
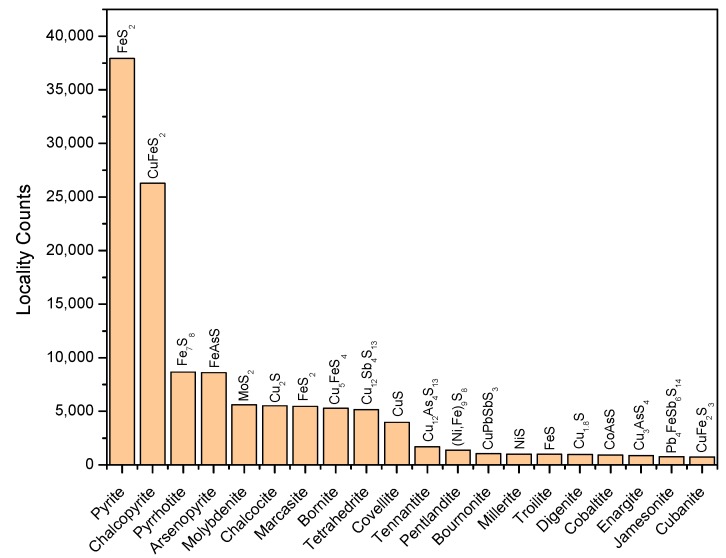
Metal sulfide distribution in natural environments. The 20 most frequently observed species are ranked in order of locality counts. The chemical composition of each species is shown.

**Figure 3 life-08-00046-f003:**
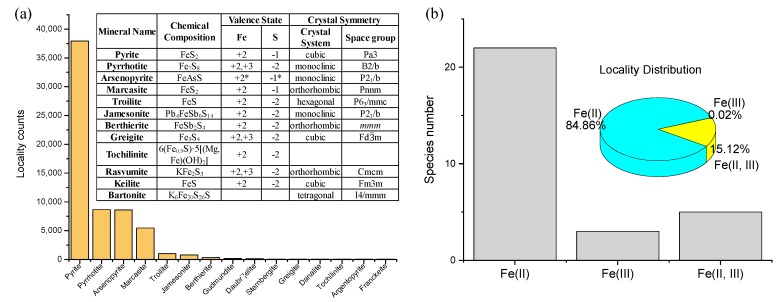
(**a**) Distribution and chemical diversity (chemical composition, Fe/S valence states, and crystal symmetry) of Fe single-metal sulfides. (**b**) Relative abundances and locality distribution of Fe^2+^-, Fe^3+^-, and Fe^2+^ plus Fe^3+^-sulfides. Note that the valence state of arsenopyrite (Fe^3+^(AsS)^3^^−^) in the RRUFF mineral database is examined as (Fe^2+^(AsS)^2^^−^) in this review based on electric spectroscopic observations [76,77].

**Figure 4 life-08-00046-f004:**
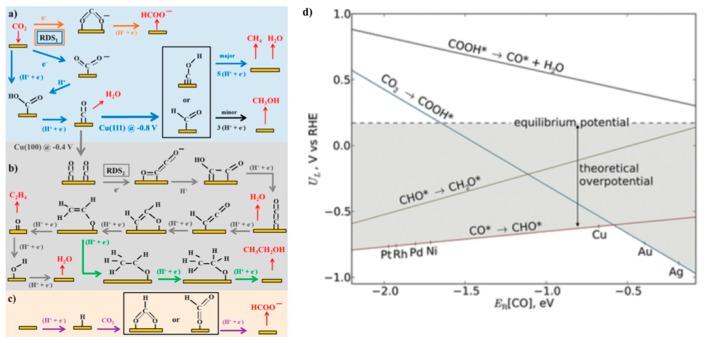
(**a**–**c**) Possible reaction pathways for the electrocatalytic reduction of CO_2_ to products on transition metals and molecular catalysts (adapted from Reference [91]): (**a**) reaction pathways from CO_2_ to CO, CH_4_ (blue arrows), CH_3_OH (black arrows), and HCOO^−^ (orange arrows); (**b**) reaction pathways from CO_2_ to ethylene (gray arrows) and ethanol (green arrows); (**c**) reaction pathway of CO_2_ insertion into a metal−H bond yielding formate (purple arrows). Species in black are adsorbates, whereas those in red are reactants or products in solution. Potentials are reported versus RHE, RDS indicates the rate-determining step, and (H^+^ + e^−^) indicates steps in which either concerted or separated proton−electron transfer takes place. (**d**) Limiting potentials (U_L_) for elementary proton-transfer steps in the reduction of CO_2_ to CH_4_ (adapted from Reference [96]). Each line represents the calculated potential at which the indicated elementary reaction step is neutral with respect to free energy and as a function of the carbon monoxide affinity (E_B_[CO]) of the electrocatalyst. The theoretical overpotential is defined as the potential difference between the most-negative limiting potential line and the equilibrium potential for the reduction of CO_2_ to CH_4_ (+0.17 V versus RHE), as highlighted in gray.

**Figure 5 life-08-00046-f005:**
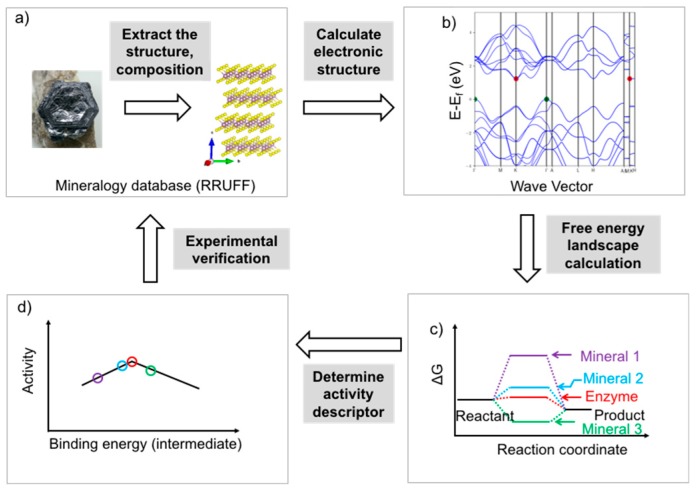
Framework of the methodology proposed for the rational screening of mineral catalysts using DFT calculations. (**a**) Select a series of minerals and extract the structure and chemical composition information from the RRUFF mineralogy database (the mineral shown here is molybdenite, MoS_2_, and was obtained from the mindat website: https://www.mindat.org/search.php?name=Molybdenite; the crystal structure was created using BIOVIA Draw software); (**b**) Calculate the electronic structure of the bulk catalyst and surface, and obtain information of the band gap, Fermi level, conductivity, and other electronic properties. The band structure shown here was obtained from the Materials Project database for molybdenite: https://materialsproject.org/; (**c**) Calculate the free energy landscape for the specific reaction (here, the reaction pathway scheme assumes that only one intermediate is involved); (**d**) Determine the activity descriptor (binding energy of the key intermediate). Using this approach, the relative activity profile for a series of minerals can be obtained. The predicted activity profile can be verified using experimental methods to validate the optimized computational model and obtain more precise structure-activity relationships.

**Figure 6 life-08-00046-f006:**
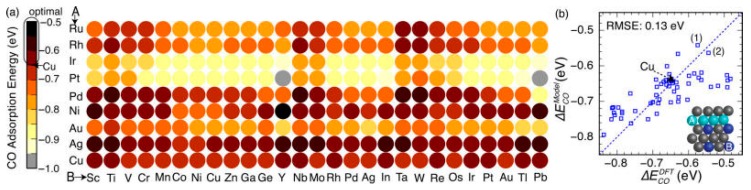
(**a**) Rational screening of CO adsorption energy on the second-generation core-shell alloy surfaces using the developed neural network model. (**b**) The parity plot shows a comparison of the CO adsorption energies on selected Cu monolayer alloys calculated by the neural network model and self-consistent density functional theory (DFT) (adapted from Reference [160]).

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
