# Peer review of "Chemical Diversity of Metal Sulfide Minerals and Its Implications for the Origin of Life"

_life, 2018, doi:10.3390/life8040046_

Round 1

Reviewer 1 Report

The review by Li et al fills a much needed gap in the prebiotic chemistry literature, related to the role of sulfide minerals in abiogenesis.  Despite the high relevance and widespread availability of these minerals they have so far been largely neglected in synthetic prebiotic chemistry.  The paper outlines several areas where catalysis by sulfide minerals could be exploited in the synthesis of biogenic molecules.    

The paper is comprehensive and well-written, it is in general a very interesting and inspiring reading.  My only objection is that perhaps the authors overemphasize a bit the importance of deep-sea hydrothermal vent scenarios of the origin of life.  I agree that the role of sulfidic minerals has most frequently been studied in this context, but there exist also scenarios which are more relevant to dry-land hydrothermal fields, which are not mentioned in the paper.  In particular, it would be worth mentioning the Zn-world hypothesis of Armen Mulkidjanian, which postulates that ZnS could play an important role in the metabolic processes leading to the emergence of life.  Further, I suggest that the authors mention the studies of the Saladino and Di Mauro groups on the role of sulfide minerals in the formamide-based synthesis of nucleic acid precursors.

Author Response

ReplyWe agree that some literatures related to the scenario in dry-land hydrothermal fields should be cited to make the introduction section complete. We have added the literatures on Zn-world hypothesis of Armen Mulkidjanian, and the ones on sulfide mineral mediated formamide conversion.  And the following sentences were newly added.

Page 4, lines 189-196

Metal sulfides can also facilitate the condensation of formamide to purine and pyrimidine nucleic bases. Saladino et al. [68] conducted a heating experiment of formamide in the presence of FeS and (Fe,Cu)S minerals as catalysts at 160 oC for 48 h. In contrast to purine as the only product in the absence of any catalyst, several types of precursors of primary metabolism, such as purine and adenine, and of nucleobase isomers, such as isocytosine, 2-aminopurine, and 2(1H)-pyrimidinone, were synthesized. The presence of iron was a crucial factor for the synthesis of purine derivatives. Sulfides with both Fe and Cu, including chalcopyrite FeCuS2 and tetrahedrite (Fe,Cu,Sb)S, show enhanced yield towards isocytosine and 2(1H)-pyrimidinone products.

Reviewer 2 Report

This review provides an interesting perspective on metal sulphides, and how their potential for electrocatalysis may be screened in the context of prebiotic chemistry. It does so by first summarizing different studies where metal sulphide catalysts have been used for prebiotically relevant reactions, such as CO2 and nitrate/nitrite reduction. Electrocatalysis is envisioned as taking place in deep sea hydrothermal vents rich in transition metals and sulphide deposits, which would provide pH, electric potential, and temperature gradients to drive carbon and nitrogen fixation. The authors present an interesting approach and analysis based on metal sulphide frequency (locality), metal composition and valence state. One assumption is that a diversity of valence states may aid the generation complex chemical networks that can lead to autocatalytic reactions. The first parts of this manuscript, sections 1-4, presents interesting ideas and summarizes important data, and will be a valuable resource for research along these lines. 

The last section (5) of the manuscript tries to propose a framework to screen for prebiotically relevant metal sulphide catalysts. The strategy proposed is a combination of activity prediction utilizing a the PDB structure library, for comparison to catalytic metal centres in enzymes and experimental verification. It is also based on extracting an adsorption descriptor from DFT calculations, mined in a structure database, for predicting catalytic activity. This part of the text seems to me more like a rewritten funding proposal than a scientific article or review. Figure 5 exemplifies this clearly: it shows what could be done, and this process is called ”rational screening”. Screening somehow implies a more or less rapid or structured process. However, it would seem to me that the suggested procedure would run into trouble at point c, which states: “calculate the free energy landscape for the specific reaction”. That is a massive and highly complex undertaking, and one that is not easily automated. There is much that could be said here. One question (out of many possible) is how do you propose to find and explore the reaction mechanism? Also point d, “determine the activity descriptor” is unclear. How would this approach be validated sufficiently to motivate the use of some descriptor outside the “training set”?   

If catalytic mechanisms could so easily be understood and projected down to a few descriptors, as this “proposed screening framework” makes it seem, then the field of catalysis would be… well, much simpler than what it really is! 

I have no objections to sections 1-4, which are well written, and seemingly relevant. However, section 5 of this manuscript is odd to me in that it suggests work to be done, instead of presenting actual work. As I question the degree of plausibility (see above), I wonder – why is section 5 in the article? My suggestion is to either do the work suggested, or some part of it, and publish that, or focus on the review aspects alone. Or, if the “proposal format” is tolerated by the editors of this journal, to restructure this part and be more clear and specific about the approach, including challenges and drawbacks. 

Below are some minor comments:

·     The English is very good. The structure is logical and easy to follow. 

·     Should there be a reference for statement on p3 row 127?

·     Figure 4a-c as is appears a little bit cluttered and could become a bit clearer with a bigger size for example. 

·     What is “deep-time analysis” on page 7?

·     Maybe some citations to and discussion  on “structure-function relationships” when this is first mentioned on page 10?

·     Just before the beginning of section 4.3, there is a very long sentence that should be split up. 

·     Is DFT (the acronym) explained when first used? 

·     Row 604 shows multi misspelled as mutli.

·     Why is “adsorption energy” within quotation marks on row 620?

·     Which are the “advances in catalytic screening” referred to on page 17?

Author Response

As to the comments of Reviewer #2

Comment:

One question (out of many possible) is how do you propose to find and explore the reaction mechanism? Also point d, “determine the activity descriptor” is unclear. How would this approach be validated sufficiently to motivate the use of some descriptor outside the “training set”? If catalytic mechanisms could so easily be understood and projected down to a few descriptors, as this “proposed screening framework” makes it seem, then the field of catalysis would be… well, much simpler than what it really is! I have no objections to sections 1-4, which are well written, and seemingly relevant. However, section 5 of this manuscript is odd to me in that it suggests work to be done, instead of presenting actual work. As I question the degree of plausibility (see above), I wonder – why is section 5 in the article? My suggestion is to either do the work suggested, or some part of it, and publish that, or focus on the review aspects alone. Or, if the “proposal format” is tolerated by the editors of this journal, to restructure this part and be more clear and specific about the approach, including challenges and drawbacks.

Reply:

According to the main claim about “However, section 5 of this manuscript is odd to me in that it suggests work to be done, instead of presenting actual work”, we referred recently published 4 papers about machine-leaning methods for rapid and accurate extraction of descriptor and thus prediction for better new catalysts. And the following new sentences were added on page 15 line 833 to page 681. We also added a new figure as Figure 6 (from ref. 165), which visually explains the concept of machine-learning-based prediction for electrocatalysts for CO2 reduction.

“It is to be noted here that the aforementioned in-silico approaches become a more exhaustive search for new catalysts owing to the integration to machine learning methods. For example, Ma et al. developed a machine-learning-augmented chemisorption model that enables fast and accurate prediction of CO2 reduction activity of metal alloys [165](Figure 6). With this model and the d-band theory, they identified a promising new catalysts for selective reduction of CO2 to C2 species [165]. Ulissi et al developed a neural-network-based surrogate to share information between activity estimates with an order of magnitude fewer explicit DFT calculations. This approach allows for an order of magnitude reduction in the number of DFT calculations required and thus rapid prediction of catalysts even for highly complicated bimetallic electrocatalyst for CO2 reduction [166]. Besides the use of in-silica data for machine learning, a recent study by Shao-Horn and coworkers [167] showed the potential to utilize experimental data from various literature for catalyst prediction after standardizing with the activity of a material common in all studies. This approach can be considered as a technical breakthrough in this field, because one of the difficulties hindering catalyst informatics was the lack of data in a uniform structure. The implementation of an internal standard allows access to the wealth of experimental data accumulated to this day, which may lead to a more widespread application of machine-learning techniques. More recently, Ooka et al developed the method to utilize genetic sequence information for catalyst development [168]. In contrast to experimentally-determined activity which deviates depending on the evaluation method (current vs overpotential, or pH conditions), the unambiguity of the gene sequence allows for a more robust statistical analysis.”

Minor comments:

Comment: Should there be a reference for statement on p3 row 127?

Reply: We added a reference on this sentence.

Comment: Figure 4a-c as is appears a little bit cluttered and could become a bit clearer with a bigger size for example.

Reply: This figure was taken from literature (citation # 92). When the paper is accepted for publication, we will replace it by higher resolution one.

Comment: What is “deep-time analysis” on page 7?

Reply: we deleted the term ‘deep-time analysis “from the revised manuscript.

Comment: Maybe some citations to and discussion on “structure-function relationships” when this is first mentioned on page 10?

Reply

Instead of adding citation, we added the following sentence to explain it.

“i.e. how the structure of active sites dictates the overall performance of catalysts”

Comment: Just before the beginning of section 4.3, there is a very long sentence that should be split up.

Reply: Sentence was split by two.

Comment: Is DFT (the acronym) explained when first used?

Reply: Yes, we did on page 14, line 482.

Comment: Row 604 shows multi misspelled as mutli.

Reply: Spell was collected to multi.

Comment: Why is “adsorption energy” within quotation marks on row 620?

Reply:  Quotation marks were removed in the revised manuscript.

Comment: Which are the “advances in catalytic screening” referred to on page 17?

Reply: We revised the sentence as follows:

Despite the recent advancement of catalytic screening mentioned above”
